# Vessel noise levels drive behavioural responses of humpback whales with implications for whale-watching

Kate R Sprogis[1,2]*, Simone Videsen[1], Peter T Madsen[1]

[1]Zoophysiology, Department of Bioscience, Aarhus University, Aarhus, Denmark; [2]Harry Butler Institute, School of Veterinary and Life Sciences, Murdoch University, Perth, Australia

**Abstract** Disturbance from whale-watching can cause significant behavioural changes with fitness consequences for targeted whale populations. However, the sensory stimuli triggering these responses are unknown, preventing effective mitigation. Here, we test the hypothesis that vessel noise level is a driver of disturbance, using humpback whales (*Megaptera novaeangliae*) as a model species. We conducted controlled exposure experiments (*n* = 42) on resting mother-calf pairs on a resting ground off Australia, by simulating whale-watch scenarios with a research vessel (range 100 m, speed 1.5 knts) playing back vessel noise at control/low (124/148 dB), medium (160 dB) or high (172 dB) low frequency-weighted source levels (re 1 μPa RMS@1 m). Compared to control/low treatments, during high noise playbacks the mother's proportion of time resting decreased by 30%, respiration rate doubled and swim speed increased by 37%. We therefore conclude that vessel noise is an adequate driver of behavioural disturbance in whales and that regulations to mitigate the impact of whale-watching should include noise emission standards.

## Introduction

Whale-watching comprises the largest component of marine mammal-based tourism, and the multi-billion-dollar industry is increasing globally (*Hoyt, 2018*). The most common form of whale-watching is boat-based, where tours often repeatedly target specific cetacean populations in easily accessible coastal waters. With an increase in the number of people whale-watching, there is consequently a rise in the number and/or the size of vessels to accommodate for the expansion of the industry.

Commercial whale-watching began in 1955 and has previously been viewed as a non-invasive activity that can generate a revenue for local economies as an alternative to whaling whilst allowing depleted whale stocks to recover. It is, however, increasingly clear that boat-based whale-watching can have short-term behavioural impacts on individuals (*New et al., 2015*). Short-term behavioural impacts include alterations of dive patterns, swim speeds, swim orientation, group cohesiveness, behavioural state and changes in acoustic behaviour (for reviews see *Senigaglia et al., 2016*; *Machernis et al., 2018*). Repeated behavioural disruptions on individuals can lead to long-term negative effects on health (i.e. body condition), survival and reproduction, which in turn, if a sufficient number of individuals are affected, can negatively influence population dynamics (*Bejder et al., 2006*; *Lusseau et al., 2006*). However, while whale-watching has been shown to have both short- and long-term negative effects on cetaceans, the sensory drivers that elicit these behavioural responses remain unclear, preventing informed mitigation.

To mitigate such negative effects and to facilitate the sustainability of the whale-watch industry, there is an increasing push by regulators for best-practice regulations, guidelines or codes of conduct. By stipulating approach distance (typically ~100 m), angle (typically from the rear and side), speed (typically below wake speed), current guidelines rest on the premise that physical proximity is

*For correspondence:
kate.sprogis@bio.au.dk

**Competing interests:** The authors declare that no competing interests exist.

**eLife digest** Whale-watching is a multi-billion-dollar industry that is growing around the world. Typically, tour operators use boats to transport tourists into coastal waters to see groups of whales, dolphins or porpoises. There is, however, accumulating evidence that boat-based whale-watching negatively affects the way these animals behave and so many countries have put guidelines in place to mitigate activities that may disturb the animals. These guidelines generally stipulate the boat's angle of approach, how close the boat can get and the speed at which it can pass by the animals.

In general, these guidelines are based on the assumption that the animals are disturbed by the closeness of the whale-watching boats. However, whales, dolphins and porpoises have very sensitive hearing, and only have a short range of vision underwater. Therefore, it seems plausible that the animals hear whale-watching boats long before they see them and so the loudness of underwater noise from the boats may be enough to disturb these animals' behaviour.

To test this hypothesis, Sprogis et al. performed experiments where they simulated a whale-watching vessel approaching humpback whale mothers and calves who were resting off the northwest coast of Australia. A small motorised research boat travelling at a low speed passed different mother-calf pairs at a target distance of 100 meters, which is a common whale-watching distance guideline in many countries. The boat had an underwater speaker that played recordings of the boat noise at different volumes, while a drone with a video camera flew overhead to record the whales' behaviours in detail and to identify individual animals.

These "controlled exposure experiments" showed that the quiet boat noise did not appear to disturb the mothers and calves. However, compared to when the quiet boat passed the animals the louder boat noise decreased how long the mother whale rested on the surface by 30%, made her swim 37% faster, and doubled the number of breaths she took per minute. If there are many disturbances from humans, then it can negatively impact the energy the mother and calf have available for nursing, fending off males and predators, and migrating back to their feeding ground nearer the Earth's poles.

Based on these findings, it is shown that the loudness of the underwater noise from boats can explain why whales may be disturbed during whale-watching activities. To help reduce this disturbance, Sprogis et al. recommend that noise emission standards should be added to the current whale-watching regulations such that boats should be as quiet as possible and ideally around the volume of the ambient background noise. This would allow operators to approach the animals in a responsible, sustainable manner and offer tourists a view of undisturbed wildlife.

the primary vehicle of disturbance (*Higham et al., 2014*). As such, a very quiet whale-watch vessel is considered to have the same impact on the target animals as a very noisy vessel at the same distance, angle of approach and speed. Given that whales in most waters of the world are offered no underwater visual cues at a range of 100 m, it therefore seems plausible that hearing rather than vision is the sensory modality that serves to mediate behavioural responses to approaching whale-watch vessels. Thus, a fundamental knowledge gap remains, as to whether behavioural reactions of cetaceans to whale-watch activities are attributable to noise level, visual cues, or a combination of both.

To alleviate that pertinent data gap, we test the hypothesis that underwater noise level from whale-watch vessels is an adequate stimulus that elicits short-term behavioural responses in whales. To do this, we measured behavioural responses of 42 humpback whale (*Megaptera novaeangliae*) mother-calf pairs to different levels of vessel noise during controlled exposure experiments on a resting ground with murky waters excluding visual cues. The humpback whale was used as a model species as it is the most targeted species for whale-watching and swim-with-whale activities globally, mainly due to its cosmopolitan distribution and highly acrobatic visual displays (*Hendrix and Rose, 2014*; *Hoyt, 2018*). In keeping with the hypothesis, we predicted that high vessel noise levels would elicit greater short-term behavioural responses than medium and control levels.

# Materials and methods

## Study area and species

Fieldwork was conducted in Exmouth Gulf, Western Australia (*Figure 1—figure supplement 1*; between 21°45′–22°33′ S and 114°08′–114°40′ E), from August 1 to October 31 2018. Humpback whales enter the Gulf during their southern migration, between late August and early November. The Gulf is an important resting ground for mother and calves, that may rest and nurse for a few weeks before continuing their southern migration to their high latitude feeding grounds (*Jenner et al., 2001*; *Bestley et al., 2019*). The Western Australian whale population has increased substantially since the cessation of whaling and is estimated to comprise 20–30,000 whales, increasing 9–12.7% yr$^{-1}$ (*Bejder et al., 2016*). The Gulf is shallow, with a mean water depth of 9 m and maximum of 20 m. The soundscape in the Gulf is dominated by biological sounds, such as the continuous melody of male humpback whale song and omnipresent snapping shrimp, with minimal noise from anthropogenic activities, such as vessels (*Bejder et al., 2019*).

## Playback noise stimuli

Controlled exposure experiments (CEEs) (*Tyack et al., 2003*) were conducted from a small research vessel (Quintrex, 6 m rigid hull, 4-stroke Yamaha 100hp outboard engine) which simulated whale-watch vessel approaches (*Figure 1*). Vessel approaches consisted of a typical whale-watch approach; transiting past a logging mother-calf pair at 100 m distance at slow speed (first gear, 800 rpm/1.5 knots; *Figure 1A*). To test the hypothesis that underwater noise level from whale-watch vessels is an adequate stimulus that elicits short-term behavioural responses in whales, the only variable that we changed during replicates of CEEs was the noise level of the same vessel signature to maximise statistical power to uncover effects of level and avoid the confounds of small differences between vessel signatures. Vessel noise was played through a laptop (Acer Aspire ES 15), amplifier (Boss PM2500 monoblock), bridging transformer box (AC1424 HP) and emitted through an underwater acoustic transducer (Lubell LL1424, flat [±3 dB] frequency range 200 Hz-9 kHz). The transducer was

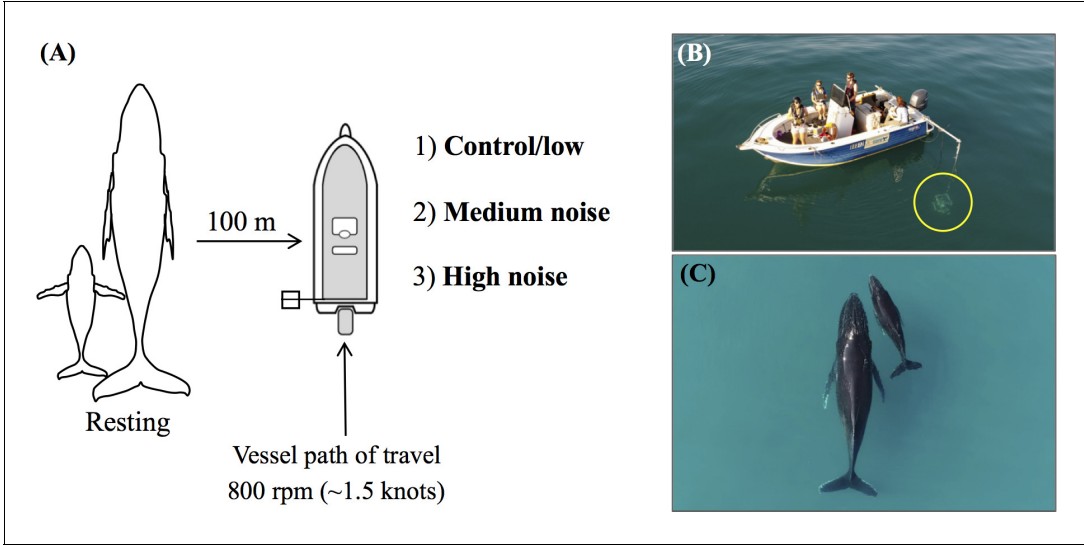

**Figure 1.** Controlled exposure experimental design to measure behavioural responses of humpback whales to different vessel noise levels. (A) Schematic showing simulated whale-watch approaches of resting mother-calf pairs in the *during* phase, at the same distance and speed, with different vessel noise playback levels (control/low 124/148 dB re 1 μPa, medium 160 dB re 1 μPa, high 172 dB re 1 μPa; not to scale). (B) Set up of the vessel and underwater acoustic transducer (circled). (C) Aerial photograph of a mother-calf pair resting on the surface, from the view of the unmanned aerial vehicle during data collection.

The online version of this article includes the following figure supplement(s) for figure 1:

**Figure supplement 1.** Study area in Exmouth Gulf, Western Australia, and research effort tracks.
**Figure supplement 2.** Spectral signatures of whale-watch vessel noise and research vessel noise.
**Figure supplement 3.** Spectral components of the vessel noise recording compared to playback noise.

suspended from the side of the vessel to 1.5 m below the surface to mimic typical depth of propellers/shaft/exhaust of whale-watching vessels (*Figure 1B*). Vessel noise was emitted with a 60 s ramp up/down to avoid an acoustic startle response from the whales. Vessel noise was set to different LF-weighted broadband source levels (SLs) of control (124 dB re 1 μPa), low (148 dB re 1 μPa), medium (160 dB re 1 μPa) and high noise (172 dB re 1 μPa RMS).

Noise files for playback were generated by recording the research vessel noise. To do this, the vessel was tied to a stationary mooring whilst the vessel was in gear at 1300–1400 rpm (equivalent of ~5 knts which is a typical speed of whale-watching vessels). Vessel noise was recorded for 60 min with an autonomous sound recorder (SoundTrap, www.oceaninstruments.co.nz). The SoundTrap was abeam at 6 m distance to engine, at 2 m depth (sampling rate of 48 kHz, 16 bit, rendering a flat (±2 dB) frequency response from 0.02 to 20 kHz, clip level 174 dB re 1 μPa (high gain)). Vessel noise was not recorded whilst in transit to eliminate flow noise from the passing water. The vessel noise recording was processed in Adobe Audition (version 3.0) using tube-modelled compression, to remove extreme values. The modified noise file was then normalised to 0.9 to amplify the sound without clipping. This modified vessel noise was used during CEEs to generate the desired SLs by changing the appropriate gain in the playback wav.files. From the 60 min noise recording, we extracted 15 different 15 min sound files at random times in the recording, for each low, medium and high noise files (i.e. 45 noise files, and a control noise). To avoid pseudoreplication, each recording was selected at random for consecutive CEEs using the function *randperm* in MATLAB (The MathWorks, Inc, Natick, MA). We deliberately chose to use a single vessel signature for noise playback to maximise the statistical power to uncover effects of noise level rather having such power diluted by the confounding effects of differences inherent in the spectral signatures of different whale-watch vessels. However, the broad scale applicability of our findings is supported by the fact that the spectral features of the playback noise is similar to a range of whale-watch vessels in the same frequency range (see *Figure 1—figure supplement 2*, Arranz, P. et al. unpublished data) (*Erbe, 2002*; *Jensen et al., 2009*; *Wladichuk et al., 2019*).

To achieve the high vessel noise level, maximum output noise level was tested at the Exmouth marina, where a maximum undistorted output level of 173 dB re 1μPa RMS was achieved (laptop volume 100% and amplifier input level 50%). To determine the low vessel noise level, we first measured the SL of the research vessel at a transit speed of 800 rpm (~1.5 knts). This was the selected transit speed as it was suitable to manoeuvre the vessel with the transducer in the water and was slow enough to replicate a whale-watch scenario (i.e. slow speed and longer duration in the presence of the whales). Source level of the research vessel was recorded by driving past a SoundTrap suspended on a weighted buoy at 5 m depth in 10 m of water at 17–18 m distance (replicated three times). The research vessel noise SL was calculated at 140 ± 2 dB re 1 μPa low frequency (LF)-weighted (*Southall et al., 2019*). We therefore had a range from 140 to 173 dB re 1 μPa to select for low, medium and high levels for CEEs, and a 12 dB difference was chosen (148, 160 and 172 dB re 1 μPa). These selected noise levels represent a range of slow moving (<10 knts) motorised whale-watch vessels that a whale would experience in the wild, having SLs ranging from 138 to 169 dB re 1 μPa @ 1 m (*Jensen et al., 2009*; *Wladichuk et al., 2019*). The control was set to 124 dB re 1 μPa to ensure it was ~16 dB quieter than the vessel noise at 800 rpm.

Prior to conducting CEEs, the accuracy of the transducer was tested at Exmouth marina, by playing the high vessel noise and recording the SL at 2 dB increments from 140 dB re 1 μPa until 173 dB re 1 μPa at 3 m depth (SoundTrap at 2 m depth, sampling rate of 48 kHz, clip level 184 dB re 1 μPa (low gain)). The SLs were linear with expected playback levels from the gain-adjusted audio files and no clipping was observed. Received levels (RLs) of playback noise were then measured in calm waters (Beaufort 0–1) in 14 m water depth using a SoundTrap attached to a vertical weighted buoy (sampling rate 48 kHz, 16 bit, clip level: 174 dB re 1 μPa (high gain)). The SoundTrap was located at 1.5 m depth from the surface as this is approximately where the lower jaw/inner ear of an adult humpback whale would be when resting at the surface. We conducted noise recordings whilst the vessel was stationary (anchored) at ~100 m distance from the SoundTrap. Ten vessel noise files were played for each low, medium and high noise treatment to confirm the LF-weighted SLs of 148, 160 and 172 dB re 1μPa (rms) (*Southall et al., 2019*; *Tougaard and Beedholm, 2019*). These LF-weighted SLs were then used to predict LF-weighted RLs by subtracting the transmission loss for each closest point of approach from the SL of the given treatment assuming spherical spreading. For verification, the RLs of vessel noise as both LF-weighted (*Southall et al., 2019*) and third-octave

level (TOLs) from the moving vessel were recorded whilst the vessel was transiting past the Sound-Trap at ~100 m distance as per CEEs (*Figure 1A*). We played four files for each noise treatment. Segments of 5 s were extracted during the closest point of approach and the different noise levels were quantified as TOLs with a 0.125 s averaging window (RMSfast) and graphed as 50th percentiles (median). *Figure 1—figure supplement 3* shows the spectral components of the playback sounds compared to the original moored vessel recording (6 m, 5 knts). It is shown that the playback noise is slightly more tonal than the recorded noise, but well within the spectral realm of whale-watching vessel spectral signatures (*Figure 1—figure supplement 2*). There is less energy at frequencies below 200 Hz in the playback noise compared to the recorded noise; however, because the noise was played to whales logging at the surface, energy at frequencies below 200–400 Hz cancel out via interference with the surface and therefore does not reach the whale (*Figure 1—figure supplement 3*). The echosounder was switched off during all recordings and experiments.

## Ambient noise

To compare the RLs of playback noise to the surrounding underwater noise in the Gulf, ambient noise was recorded. Ambient noise was recorded continuously by a SoundTrap (sample rate of 48 kHz, clip level 174 dB re 1 µPa attached to a weighted buoy in 15 m water depth (the SoundTrap was ~4 m below the surface) at 22°00'19' S 114°08'25' E (*Figure 1—figure supplement 1*) from 31 August to 23 September. Ambient noise was also recorded for 24 hr periods (18 September and 18 October). For analyses, ambient noise was quantified as TOLs in dB re 1 µPa RMS (0.125 s averaging windows). Five different relevant TOLs (nominal center frequencies of 250 Hz, 400 Hz, 1250 Hz, 2500 Hz, 4000 Hz) were extracted following *Bejder et al., 2019*. Self-noise of SoundTraps were recorded in a silent anechoic chamber at Aarhus University, Denmark.

## Controlled exposure experimental design

Controlled exposure experiments were conducted on humpback whale mother-calf pairs, where the mother was predominantly logging on the surface. Mother-calf pairs were selected as i) many whale-watch and swim-with-whale operators target them during tours due to their slow, calm behaviour (*Sprogis et al., 2020*), ii) they are likely the most sensitive to anthropogenic disturbance (*Lundquist et al., 2013*; *Argüelles et al., 2016*), and iii) they offer a standardised behaviour that facilitates detection of noise induced disturbance. Controlled exposure experiments consisted of a *before, during* and *after* experimental design (*before* phase = absence of vessel, stationary >300–400 m from whales with engine in neutral; *during* phase = vessel approach (*Figure 1*); *after* phase = departure of vessel, stationary >300 m from whales with engine in neutral). In the *during* phase, the driver of the vessel aimed for a tangential to parallel approach, and pass at the same distance (~100 m) distance and speed (~1.5 knts). To ensure the distance to the whale, a laser range finder (Bushnell 10 × 42 Fusion 1 mile laser) was used.

We aimed to conduct ten replicates of each treatment to ensure sufficient power for analyses. Replicates of CEEs were conducted >3 km from any previous CEE on the same day. To ensure samples were independent (same mother-calf pair never sampled twice), photo-identification of the dorsal fin using a DSLR (Canon 50D 400 mm lens) and aerial photographs of the dorsal side were taken using an unmanned aerial vehicle (UAV). Focal follows were conducted during good weather conditions (<15 knot winds, Beaufort sea state <3) and ceased if the weather deteriorated. Controlled exposure experiments were included in analyses if *before, during* and *after* data was recorded on the same mother-calf pair, if the research vessel approached ~100 m, if no other vessels passed <500 m of the focal pair, and if no conspecifics or other species approached and/or interacted <100 m of the focal pair.

## Unmanned aerial vehicle focal follows of mother-calf pairs during CEEs

Throughout CEEs, focal follows of humpback mother-calf pairs were conducted using UAVs to video-record continuously for all occurrence sampling (*Altmann, 1974*). A quadcopter UAV (DJI Phantom 4 Advanced, diameter = 350 mm, weight = 1368 g, video = 2.7K, 2720 × 1530, 48fps) was flown, which had a maximum flight time of ~25 min. The UAV provided a live video feed to the remote controller connected to an iPad (*Figure 1C*). The UAV was launched and retrieved by hand from the front of the vessel. Two UAVs were flown consecutively to ensure a near-continuous video

recording. UAVs were flown at 25–30 m altitude and positioned above the mother, facing north and the camera vertically down (*Nielsen et al., 2019*). At these altitudes, the presence of the UAV has no negative noise effects due to the low RLs underwater (*Christiansen et al., 2016b*), and cause no apparent behavioural changes on baleen whales (*Christiansen et al., 2020*), thus this technology is non-invasive and appropriate to record undisturbed (control) behavioural responses. The UAV logged UTC time, GPS position (WGS84 ellipsoid) and altitude (barometric and GPS) every 100 ms.

## Response variables to determine behavioural responses of whales

Behavioural responses of interest were common short-term responses that are altered during whale-watch activities (*Senigaglia et al., 2016*), namely i) behavioural events, ii) respiration rate, iii) heading and iv) swim speed.

### Behavioural events of mother and calves

Behaviours were either instantaneous (precise moment in time, e.g. peduncle dive) or continuous (the duration of time that behaviour lasted, e.g. logging as a proxy for resting) (*Altmann, 1974*). An increase in instantaneous behaviours is an indicator of disturbance to whale-watching activities (*Machernis et al., 2018*). Behavioural events of mother and calves were processed using *Solomon Coder* v17.03.22 (András Péter, https://solomon.andraspeter.com/). Behaviours were identified in UAV videos from a behavioural ethogram (*Supplementary file 1*) and registered manually in a blind review (phase/treatment unknown). The proportion of time resting (a continuous value from zero to one), consisting of the sum of logging duration divided by total duration of the phase, and presence of instantaneous events during a focal follow (presence 1, absence 0) were used in analyses.

### Respiration rate of mother and calves

Respiration rate relates to energy expenditure, for example, during whale-watch interactions the respiration rate of minke whales increased, irrespective of swim speed, indicating an increase in metabolic rate (*Christiansen et al., 2014*). Respirations were registered using in Solomon Coder, for both the mother and calf. Respiration rate was calculated as the number of breaths divided by the duration of the phase (number of breaths min$^{-1}$) and was used in analyses.

### Heading of mother

The deviation index (DEV) is a measure of movement predictability, to capture how much the whale is changing heading during its track (*Williams et al., 2002*; *Christiansen et al., 2013*), and in our case, to capture how much the whale is rotating whilst logging on the surface (e.g. to incorporate rotational movements). DEV ranges between 0° (smooth predictable movements) and 180° (erratic movements). To acquire the heading of the mother, images were extracted from the UAV video every three seconds. Stillframes were processed using a custom-made imaging program (LabVIEW vision development module v18.0.1, National Instruments). To detect the mother, the contrast of the images were adjusted so that the mother's body shape was contrasted against the background (e. g. ~0.7 threshold non-linear contrast enhancement). The program did an automatic detection test for the largest object in the frame (the mother), and values were used when the UAV camera was facing north and vertically down above the mother. Output files were audited manually, and corrections were made if the program did not detect the mother's heading correctly. Phases were excluded if <5 heading data points were obtained. The DEV of the mother for each phase was used in analyses and was calculated by obtaining the absolute heading changes between adjacent points, and taking the mean of these values (in degrees). The heading of the calf was not considered, as the calf's heading was erratic.

### Swim speed of mother

Increased swim speeds can be referred as a horizontal avoidance strategy during whale-watching activities (*Lundquist et al., 2013*). The mother's speed of movement was extracted from the UAV GPS position (lat/long). The swim speed of the calf was not considered, as the calf's swim speed was dependent on the mother. Data were only included in calculating swim speed when the camera was facing north and vertically down (−87° to −90°) above the mother. Swim speeds were then estimated between every GPS position in *R*. Swim speed data was excluded if swim speeds were unrealistic

(i.e. usually during the initial positioning of the UAV, >8 ms$^{-1}$). The mean swim speed (ms$^{-1}$), calculated by dividing the distance travelled by the duration of the focal follow video recording, was calculated for each phase. Low mean speed values (<0.1 ms$^{-1}$) were logging whales that were drifting with the tide, and these values remained in analyses so the effect of tide was representative for all focal follows. Thus, the terminology 'swim speed' represented the speed of movement for logging whales on the surface drifting with the tide, slow travelling and faster moving whales.

## Data analyses for the behavioural effects on whales

Data were filtered, and CEEs that ceased early and/or were not used in analyses were due to i) identifying repeat whales ($n$ = 2), ii) if a boat approached close by ($n$ = 5), iii) if the sun was setting and did not allow sufficient time to complete the CEE ($n$ = 1), iv) if conspecifics arrived <100 m to the focal pair ($n$ = 12), v) if the vessel ended up approaching the whale too close (e.g. 70 m, $n$ = 1), vi) if the whales possibly reacted to a loud gear-shift in the *during* phase ($n$ = 2), vii) if the whales were predominantly slow travelling in the *before* phase ($n$ = 15), viii) due to technical issues ($n$ = 4), ix) if the weather deteriorated ($n$ = 1), and x) if other species interacted with the whales appearing to cause behavioural changes (e.g. a school of fish, $n$ = 1; silver gulls pecking the skin of the whale, $n$ = 6). The remaining CEEs that had *before*, *during* and *after* data on the same mother-calf pair were used in analyses.

Mixed effect models were constructed to investigate the effects of underwater vessel noise on mother-calf pairs, and were developed in *R* v3.5.2 (*R Development Core Team, 2014*). Prior to modelling, data exploration was conducted following *Zuur et al., 2010*. We examined i) within treatments to determine if there was an effect of treatment, and ii) among treatments to determine the severity of the treatment. Treatment and phase were the fixed effects of interest, thus an interaction term between treatment and phase was added as treatment was dependent on phase. A CEE was composed of three phases (*before, during, after*) on the same focal whale (i.e. repeated measures), thus to account for any effect of individual, mother-calf identity was added as a random effect.

Five response variables were tested for within and among treatment effects: 1) the proportion of time resting, 2) presence of instantaneous events, 3) respiration rate, 4) heading change, and 5) swim speed (*Supplementary file 2* for model summary). We developed linear mixed effects models (LMMs: models 3, 4 and 5) in the *nlme* package (*Pinheiro et al., 2019*), and when data did not conform to normality we used generalised linear mixed models (GLMM: models 1 and 2) in the *MASS* package (*Venables and Ripley, 2002*) using penalized quasi-likelihood (GLMM-PQL) to account for overdispersion (*Bolker et al., 2009*) following *Zuur et al., 2009*.

To validate models, normalised residuals versus fitted values were calculated to identify potential violations of model assumptions. We explored scatterplots for homogeneity, histograms for normality, Cook's distance for influential points and auto-correlation functions (ACF plots) for temporal dependence. The goodness-of-fit for each model was assessed using a coefficient of determination ($R^2$, ranging from 0 to 1). The marginal ($R^2_{(m)}$) and conditional ($R^2_{(c)}$) values were calculated in the *MuMIn* package (*Nakagawa and Schielzeth, 2013*). $R^2_{(m)}$ explains the variance explained by the fixed effects, while $R^2_{(c)}$ explains the variance in the full model (including the random effects).

## Results

### Controlled exposure experiments and focal follow effort

Controlled exposure experiments were conducted across 53 days (290 hr on the water) from 25 August to 28 October, 2018 (*Figure 2*). Vessel track lines searching for resting mother-calf pairs covered 2337 km (*Figure 2—figure supplement 1*). Over 60 CEEs were conducted, including 273 UAV flights (78 hr of flight time). The length of mothers ranged from 9.7 m to 16.7 m (mean = 13.2 m; $n$ = 105), and for calves from 4.5 m to 8.4 m (mean = 6.3 m; $n$ = 104; *Figure 2—figure supplement 1*). After data filtering, 13 control/low (including 4 low), 14 medium and 15 high noise CEEs were used in analyses (42 mother-calf pairs; *Figure 2*; *Video 1* for examples of reactions). Control and low noise treatments were pooled as the RLs of vessel noise were around ambient noise (*Figure 3*). For these filtered CEEs, there was 29.4 hr of data conducted in daylight hours (7:20 to 18:20). The mean duration for *before* flights was 13:09 mins (0.003 SD), *during* was 15:04 min (0.002 SD) and *after* was 14:48 min (0.004 SD). The average closest point of approach was 135 m (56.0 SD). Filtered CEEs

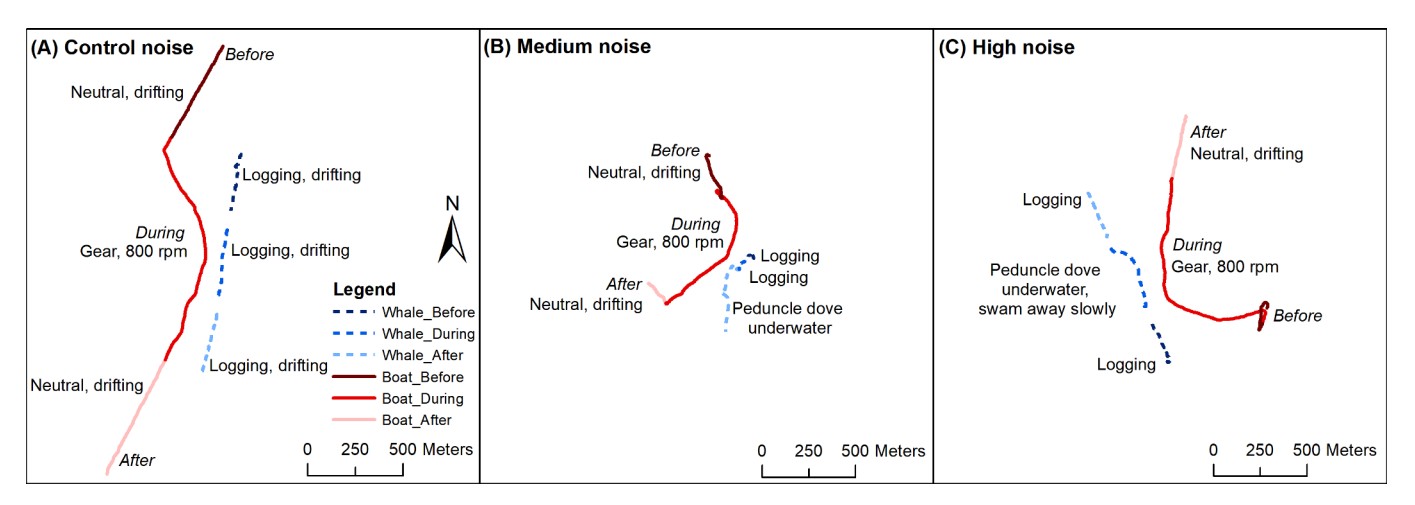

**Figure 2.** Examples of controlled exposure experiments of the research vessel simulating a whale-watch vessel approach. Three focal mother-calf pairs (dashed line) in *before*, *during* and *after* phases with the research vessel approaching (solid line) at 800 rpm (~1.5 knts) during control, medium and high noise treatments. Details of these CEEs are: **(A)** control: mother resting/logging the entire duration drifting with the incoming tide (flow from ~north to south), closest point of approach 97 m, **(B)** medium: mother logging *before*, mother logging *during*, mother peduncle dove underwater *after*, high tide, closest point of approach 95 m, **(C)** high: mother logging *before*, mother logging *during* until closest point of approach then peduncle dove underwater and swam away slowly just under the surface of the water, mother logging *after*, outgoing tide (flow from ~south to north), closest point of approach 95 m.

The online version of this article includes the following figure supplement(s) for figure 2:

**Figure supplement 1.** Absolute body length of humpback whale mother and calves.

were conducted in water temperatures between 19°C and 24°C (mean = 22°C) and in water depths between 13 m and 21 m (mean = 17 m).

## Received noise levels from the research vessel and ambient noise

From the transiting calibration, all treatments had a peak level in the third octave band around 4 kHz, with mean LF-weighted RLs @100 m of: control at 104 dB, low noise at 112 dB (±1) (average distance: 103 m, range: 102–103 m), medium noise at 122 ± 3 dB (average distance: 102 m, range: 94–108 m), and high noise at 133 ± 2 dB (average distance: 101 m, range: 85–113 m) re 1 µPa RMS (0.125 s) (*Figure 3*; *Figure 3—figure supplement 1* for shallower depth). When we correct for spherical spreading, the back calculated SLs are: medium noise at 162 ± 2 dB, and high noise at 173 ± 2 dB re 1 µPa RMS (0.125 s), in keeping with the predictions from the stationary calibrations (*Supplementary file 3*). Control/low noise was equal to the ambient noise statistic, so it was not meaningful to compute SLs from these RLs. Ambient noise was mostly dominated by humpback whale song and snapping shrimp (*Figure 3—figure supplement 2*). Median LF-weighted levels increased by a moderate 5 dB from 103 to 108 dB re 1uPa RMS (0.125 s). However, the TOLs between 400 Hz and 2500 Hz rose by 10–15 dB as singing humpback whale males arrived through the breeding season (*Figure 3*), whereas high frequency TOLs changed little due to the ever-present snapping shrimp noise.

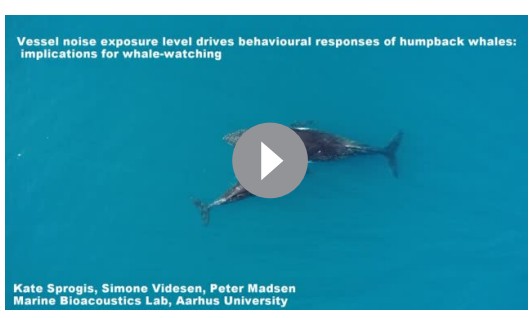

**Video 1.** Focal follow examples of a control, medium and high noise controlled exposure experiment *during* simulated vessel approaches.
https://elifesciences.org/articles/56760#video1

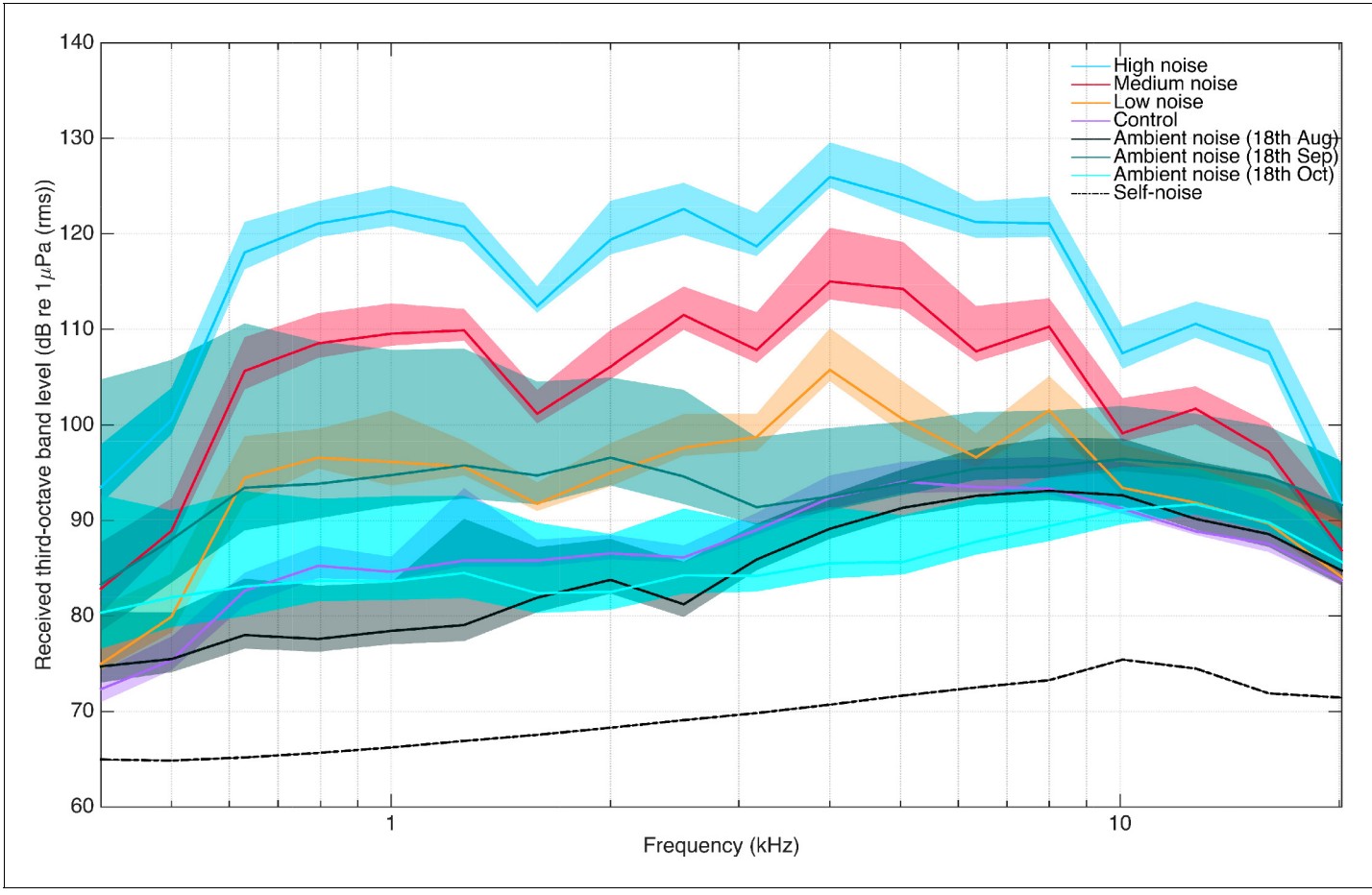

**Figure 3.** Received levels in noise treatment calibration quantified as third-octave level bands in dB re 1 μPa RMS. RLs are across third-octave band frequencies 400 Hz-20 kHz. The vessel was transiting at 800 rpm at ~100 m (range 85–113 m) distance to the SoundTrap, in 14 m water depth. 50th percentiles of control (purple line), low (yellow line), medium (red line) and high (blue line) vessel noise. Ambient noise 18th August (dark green): humpback whale male whale song absent. Ambient noise 18th September (medium green): peak whale season with song. Ambient noise 18th October (light green): towards the end of the whale season (integration bins 0.125 s). The transparent area around each percentile line corresponds to the 25th and 95th exceedance levels. Self-noise of the SoundTrap plotted as dashed black line.

The online version of this article includes the following figure supplement(s) for figure 3:

**Figure supplement 1.** Noise treatment received levels quantified as third-octave level bands (TOLs) in dB re 1 μPa RMS in 6.8 m depth.

**Figure supplement 2.** Ambient noise recorded in Exmouth Gulf, Western Australia.

## Controlled exposure experiments- behavioural responses of whales

### Continuous behavioural event- proportion of time resting for mother and calves

There were no significant effects for the proportion of time resting for the mother across phases within control/low and medium treatments (GLMM-PQL logit scale: $R^2_{(m)}$ = 0.09, $R^2_{(c)}$ = 0.22; 126 phases; *Figure 4A*, *Figure 4—figure supplement 1*). Within high noise treatments, the proportion of time resting from *before* to *during* vessel approaches decreased significantly (p=0.02,), with a 27% decrease in resting (*Figure 4A*). *After* high noise approaches, the proportion of time logging was not significantly different from *before* vessel approaches (p=0.07). Among treatments, the proportion of time resting for the mother decreased significantly from *during* control/low to *during* medium (p=0.03, 23% decrease) and *during* high (p=0.01, 30% decrease) treatments.

There were no significant effects within treatments on the proportion of time resting for the calf, for example, between *before* to *during* and *before* to *after* phases (GLMM-PQL logit scale: $R^2_{(m)}$ = 0.08, $R^2_{(c)}$ = 0.25; 126 phases; *Figure 4B*, *Figure 4—figure supplement 1*). As the vessel passed *during* high noise, similarly to the mother there was a 29% decrease in resting (although not

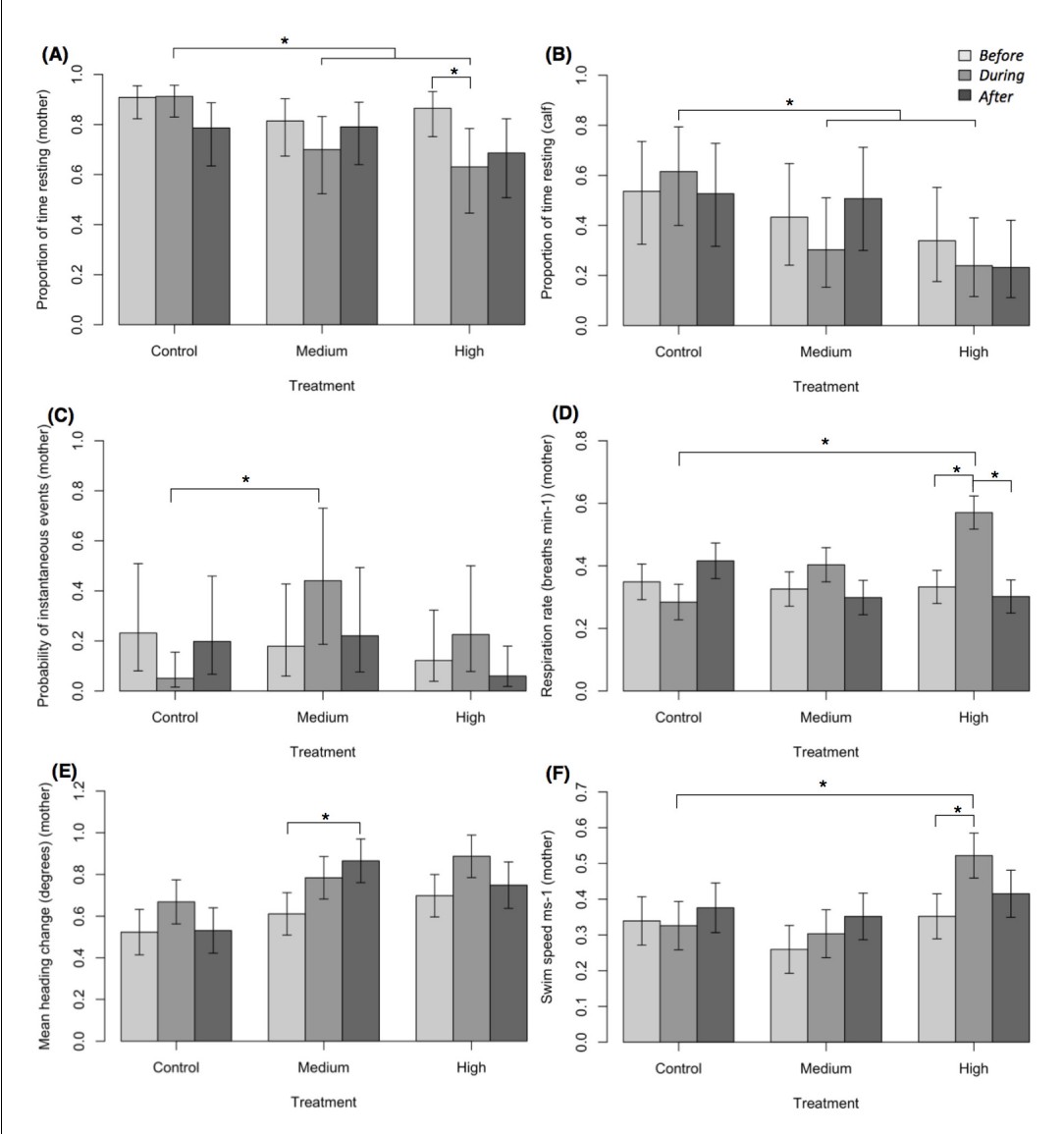

**Figure 4.** Short-term behavioural responses of humpback whales. (A) Proportion of time resting for the mother, (B) proportion of time resting for the calf, (C) probability of instantaneous behavioural events for the mother, (D) respiration rate for the mother, (E) mean heading change for the mother, and (F) mean swim speed for the mother. Representation of the model (back-transformed from the logit scale A, B, C). Vertical lines represent standard errors. Asterisks indicate significant differences among phases and/or treatments.

The online version of this article includes the following figure supplement(s) for figure 4:

**Figure supplement 1.** Coefficient plots of model outputs.

significant; *Figure 4B*). Among treatments, the proportion of time resting for the calf decreased significantly between *during* control/low to *during* medium (p=0.02, 36% decrease) and *during* high (p=0.005, 49% decrease) treatments.

## Instantaneous behavioural events for mother and calves

Across all CEEs, the occurrence of instantaneous behavioural events for the mother were limited ($n$ = 59). Events were present in 21% (26 of 126) of phases. The presence of events did not differ significantly across phases within control/low, medium and high treatments (GLMM-PQL logit scale: $R^2_{(m)}$ = 0.11, $R^2_{(c)}$ = 0.39; 126 phases; *Figure 4C*, *Figure 4—figure supplement 1*). Among

treatments, the presence of events differed significantly from *during* control/low to *during* medium treatments (p=0.02, *Figure 4C*). Events registered for the mother, included, peduncle dive, fluke dive and roll. Peduncle dives were the most common across phases (47%, 28 of 59 events), predominantly occurring *during* medium and high treatments (*during* low, 0%, 0 of 1 event; medium, 44%, 4 of 9 events; high, 73%, 8 of 11 events).

The presence of instantaneous behavioural events for the calf was greater than what was recorded for the mother (*n* = 444). Events were present in 64% (81 of 126) of phases. Events commonly registered across calves were peduncle dives and rolls. There was no significant effect of treatment, phase or their interaction on the presence of events for the calf (GLMM-PQL logit scale: $R^2_{(m)}$ = 0.05, $R^2_{(c)}$ = 0.33; 126 phases).

### Respiration rate of mother and calves

The average respiration rate for the mother *during* control/low treatments was 0.3 (±0.12 SD), *during* medium was 0.4 (±0.15 SD) and *during* high noise was 0.6 (±0.39 SD) respirations min$^{-1}$. For the respiration rate of the mother, within control/low and medium treatments, there were no significant effects across phases (LMM: $R^2_{(m)}$ = 0.15, $R^2_{(c)}$ = 0.46; 126 phases; *Figure 4D*, *Figure 4—figure supplement 1*). Within the high treatments, the respiration rate from *before* to *during* vessel approaches increased significantly by 42% (p=0.001, *Figure 4D*). Among treatments, respiration rate *during* high noise was twice that of *during* control/low noise (p = <0.001; *Figure 4D*).

The average respiration rate for the calf in *before* phases was 0.9 (±0.42 SD), 1.3 (±0.46 SD) and 1.3 (±0.65 SD) and in *during* phases was 1.1 (±0.37 SD), 1.3 (±0.23 SD) and 1.3 (±0.61 SD) for control/low, medium and high noise treatments, respectively. There were no significant effects on the respiration rate of calves (LMM: $R^2_{(m)}$ = 0.06, $R^2_{(c)}$ = 0.42; 126 phases).

### Heading of mother

There were no significant effects for the mean heading of the mother (DEV) among phases or within control/low and high noise treatments (LMM: $R^2_{(m)}$ = 0.10, $R^2_{(c)}$ = 0.50; 119 phases [*before* = 40, *during* = 41, *after* = 36]; *Figure 4E*, *Figure 4—figure supplement 1*). Within medium noise, the mean heading change from *before* to *after* vessel approaches increased significantly by 29% (p=0.02). Among treatments, heading did not differ significantly from *during* control/low to *during* medium or high noise.

### Swim speed of mother

The mean speed of movement for the mother was 0.3 ms$^{-1}$ (±0.24 SD) *during* control/low, 0.3 ms$^{-1}$ (±0.17 SD) *during* medium and 0.5 ms$^{-1}$ (±0.36 SD) *during* high noise. For the swim speed of the mother, within control/low and medium treatments, there was no significant differences across phases (LMM: $R^2_{(m)}$ = 0.08, $R^2_{(c)}$ = 0.57; 121 phases [*before* = 41, *during* = 40, *after* = 40]; *Figure 4F*, *Figure 4—figure supplement 1*). Within high noise treatments, swim speed increased significantly by 33% from *before* to *during* vessel approaches (p=0.007). Among treatments, swim speed *during* high noise increased significantly compared to *during* control/low treatments (p=0.04, 37% increase).

## Discussion

Whale-watching activities are growing rapidly globally, and the consequences of repeated behavioural disruptions are of concern both in terms of the fitness of individuals and populations (*Bejder et al., 2006*; *Lusseau et al., 2006*). Although identification of the disturbance cues (e.g. visual and/or auditory) from whale-watching activities is critical for informing relevant mitigation measures and operational guidelines, there are currently, to our knowledge, no data offering direct tests of the sensory vehicles of disturbance from whale-watching operations. Through controlled exposure experiments, we show that underwater noise level from a vessel, representative of motorised whale-watch vessels (*Figure 1—figure supplement 2*), is an adequate stimulus evoking disturbance from whale-watching on resting humpback whale mother-calf pairs on a resting ground.

Within control/low noise treatments, the maximum LF-weighted RL of vessel noise was 112 ± 1 dB re 1 µPa RMS, which was close to the ambient noise levels in Exmouth Gulf. In accordance, there

were no detectable behavioural responses in mother-calf pairs to this quiet vessel noise (*Figures 3* and *4*). Ambient noise levels increased during the season with the arrival of singing humpback whale males, however, the high and medium vessel noise CEE were very likely still audible to the whales (*Figure 3*). Within medium noise treatments, with a maximum LF-weighted RL of 122 ± 3 dB re 1 µPa RMS, the mother's behaviour was significantly affected *after* vessel approaches as her mean heading change increased by 29% from *before* approaches. Within high noise treatments, with a maximum LF-weighted of RL 133 ± 2 dB re 1 µPa RMS, the mother's behaviour was significantly affected from *before* to *during* vessel approaches as her amount of time resting decreased by 27%, respiration rate increased by 35% and swim speed increased by 33%. Furthermore, the severity of behavioural responses differed among treatments, with louder noise exposures in *during* phases causing more severe behavioural responses in mothers (*Figure 4*). These behavioural responses represent avoidance strategies and an increase in energy expenditure, which are comparable to other studies that examine the effects of whale-watching on cetaceans (*Senigaglia et al., 2016*; *Machernis et al., 2018*). Thus, we conclusively demonstrate that higher noise levels from the same vessel at the same approach distance and speed evoke significant short-term behavioural responses from humpback whale mother-calf pairs. Conversely, low noise levels from the same vessel do not lead to any significant detectable behavioural effects.

## Implications for mother-calf energy budgets and fitness

Humpback whale mothers rest on average for 35% of their time on the breeding ground as the early phase of lactation is the most energetically demanding phase in their reproductive cycle, with females sometimes loosing >25% of their body condition in only 3–4 months (*Christiansen et al., 2016a*; *Bejder et al., 2019*). Thus, an increase in respiration rate and movements due to anthropogenic disturbance (e.g. whale-watching with a loud vessel) will increase maternal energy expenditure, especially if loud noise exposures are cumulative (e.g. repeated throughout the day, from many sources, or prolonged exposure). Such noise-induced disturbances inevitably leads to a negative offset in the energy available for nursing, fending off males/predators and migrating back to their polar feeding ground (*Braithwaite et al., 2015*). For calves, there was no significant effect of playbacks on instantaneous events or respiration rate; however, the proportion of time resting decreased with increased vessel noise level. As the mothers were significantly disturbed during high noise playbacks, and as the calf (<3–4 months old) is dependent on its mother, the energetic consequences are also likely to increase for the calf. A calf is required to nurse substantially (~20% of their time; *Videsen et al., 2017*) to grow in strength and size (by ~3 cm in length a day; *Christiansen et al., 2016a*) within a short period of time to reduce predation (killer whales and sharks; *Pitman et al., 2015*) and endure the long migration to high latitude feeding grounds (*Bestley et al., 2019*). Thus, disturbing mother-calf pairs on a resting ground through loud vessel noise should be minimised and mitigated appropriately, especially as the whale-watch industry is expanding globally. We show that with a quiet vessel, mothers continued to rest, thus in a whale-watch scenario mothers and calves are more likely to remain at the surface which is beneficial for tourist viewing compared to if the mother was to dive and swim away as documented during medium and high vessel noise playbacks.

## Implications for management of whale-watching

The demonstration that noise exposure level from a vessel is a driver of short-term behavioural disturbance in whales is perhaps not surprising, as for cetaceans hearing is the primary sensory modality which is more efficient than sight at longer ranges (*Richardson et al., 1995*). Cetacean sight underwater is highly limited and hence even under the best conditions they are unlikely to be able to see a vessel underwater at 100 m distance. Thus, to facilitate sustainable whale-watching operations, guidelines and legislation should be based on approach distance, angle, speed and as highlighted here noise level. For humpback whales, we provide an evidence-based noise threshold for recommended application into global whale-watch guidelines: for vessels with spectral signatures similar to the playback used here, we suggest that emission standards be implemented so that vessels operating around whales as close as 100 m do not have LF-weighted SLs of more than 150 dB re 1 µPa RMS @ 1 m. Such low vessel SLs will lead to RLs for a logging whale at 100 m distance that are close to the ambient noise (depending on habitat, sea state and timing in the whale season), offering noise levels that are perhaps audible to the whales but with a low perceived loudness. Clearly, it

remains an open question of how this noise limit may apply to other cetacean species with different predator avoidance strategies, behavioural states, exposure history, habitats and hearing capabilities. Nevertheless, as noise from non-cavitating vessels has most of the energy at low frequencies (*Jensen et al., 2009*), it seems parsimonious to hypothesise that smaller toothed whales would similarly not respond to whale-watching vessels with broadband SLs <150 dB re 1µPa RMS at 100 m given their poor low frequency hearing (*Au et al., 1997*). To further examine the impacts of vessel noise, we suggest that future research should address the noise effects from the presence of multiple vessels, different vessel engine types, vessel proximity and different vessel approach types (e.g. in-path vs. parallel approaches during whale-watching/swim-with-cetacean tourism).

There is currently limited information on the range of SLs produced by commercial whale-watch vessels around whales. Available data suggests a SL range from around 138 to 169 dB re 1 µPa @ 1 m for slow moving (<10 knts) whale-watch vessels (*Au and Green, 2000*; *Jensen et al., 2009*; *Wladichuk et al., 2019*), largely covering the low to high range of playback SLs used here. Similarly, the chosen playback noise has spectral features representative of signatures of whale-watch vessels (*Figure 1—figure supplement 2*) and other motorised vessels (*Erbe, 2002*; *Jensen et al., 2009*; *Wladichuk et al., 2019*). Thus, if these spectral signatures and SLs are representative, some vessels are already at the low output levels recommended here when moving slowly, while others produce higher SLs that are expected to evoke clear responses at 100 m from humpback whales similar to those documented here. For loud vessels, perhaps most importantly, operators can reduce the SL of their vessel by driving slowly (*Erbe, 2002*; *Jensen et al., 2009*; *Wladichuk et al., 2019*), avoid gear-shifts which generate high-level transient sounds (*Jensen et al., 2009*), and/or increase the distance to the focal whales, although just a 6 dB increase in SL would require a doubling in range to 200 m, which tour operators are unlikely to do. To permanently reduce SLs, operators can apply a range of techniques, including using larger, slower moving propellers to minimise cavitation, quieter engines/ electric engines and installing noise absorption gear (*International Maritime Organisation, 2014*).

## Conclusion

This study demonstrates that noise level from a vessel drives the short-term behavioural response of humpback whales to disturbance. Thus, in the commercial whale-watch industry, if two operators adhere to distance guidelines (e.g. 100 m distance to the whale) with two different vessels with a similar spectral signature as our playback (one quiet and one loud), the vessels would evoke vastly different responses from the whales, resulting in differences in energetic and fitness consequences. To reduce or avoid disturbance, we propose that noise emission standards be incorporated into whale-watch and swim-with-cetacean guidelines along with the already stipulated speed, distance and angle of approach regulations. We recommend that whale-watch vessels employ broadband SL <150 dB re 1 µPa RMS when operating around whales at guideline distances of 100 m. Given that some vessels already comply with this, these recommendations are feasible to implement into existing whale-watch guidelines. These recommendations will allow operators to approach cetaceans in a responsible, sustainable manner in a way that also offers eco-tourists a view of undisturbed wildlife.

## Acknowledgements

We greatly appreciate the field, office and data processing assistance of P Harkness, J Groeneveld, M Nielsen, N Maurer and J Schilling. We thank M Johnson for advice on experimental design and K Beedholm for assistance with playback files and development of the Labview program. We are grateful to D Sprogis for electronic assistance and J Jensen for field equipment development. Thank you to F Christiansen and L Rojano-Doñate for statistical input. We are appreciative for comments on the manuscript from F Christiansen and G Minton. We thank Interspacial Aviation Services Pty Ltd for UAV operational support and training. Thank you to the reviewers and P Miller for strengthening the manuscript. This project received funding from the European Union's Horizon 2020 research and innovation programme under the Marie Skłodowska-Curie grant agreement No 792880 to KS. Additional field research funds were provided by the Independent Research Fund Denmark (FNU) frame grants to PM.

## Additional information

### Funding

| Funder | Grant reference number | Author |
|---|---|---|
| H2020 Marie Skłodowska-Curie Actions | 792880 | Kate R Sprogis |
| Natur og Univers, Det Frie Forskningsråd | 6108-00355B | Peter T Madsen |
| Independent Research Fund Denmark | | Peter T Madsen |

The funders had no role in study design, data collection and interpretation, or the decision to submit the work for publication.

### Author contributions

Kate R Sprogis, Conceptualization, Data curation, Formal analysis, Funding acquisition, Validation, Investigation, Visualization, Methodology, Writing - original draft, Project administration, Writing - review and editing; Simone Videsen, Formal analysis, Investigation, Methodology, Writing - review and editing; Peter T Madsen, Conceptualization, Resources, Supervision, Funding acquisition, Visualization, Methodology, Writing - review and editing

### Author ORCIDs

Kate R Sprogis ⓘ https://orcid.org/0000-0002-9050-3028
Simone Videsen ⓘ http://orcid.org/0000-0002-7563-2470
Peter T Madsen ⓘ http://orcid.org/0000-0002-5208-5259

### Ethics

This study was carried out with approval from the Murdoch University Animal Ethics committee (R3048/18) and was licensed by the Western Australian Department of Biodiversity, Conservation and Attractions (08-002407-3). UAV operations were conducted under a UAV Operator's Certificate (CASA.ReOC.0075) and a remotely piloted aircraft system licences (K. Sprogis) in accordance with regulations by the Australian Civil Aviation Safety Authority.

### Decision letter and Author response

Decision letter https://doi.org/10.7554/eLife.56760.sa1
Author response https://doi.org/10.7554/eLife.56760.sa2

## Additional files

### Supplementary files

• Supplementary file 1. Humpback whale behavioural ethogram. Ethogram for surface behavioural events and conspicuous underwater events, for both instantaneous and continuous events, on a resting ground.

• Supplementary file 2. Mixed models used in analyses to test for behavioural effects of underwater vessel noise on humpback whales. Models were linear mixed effects models (LMM) and penalized quasi-likelihood generalised liner mixed models (GLMM-PQL). Interaction of fixed effects = treatment*phase. Random effect = (1|Individual). Weights = the duration of time for each phase e.g. more weight will be given to longer phases. Corr = to account for temporal auto-correlation within follows, the model was used with an auto-regressive structure with lag one. REML = restricted maximum likelihood estimation. † Fitted for both mother and calf.

• Supplementary file 3. Perceived received levels of vessel noise by focal whales. Broadband source levels (SL) of vessel noise treatments produced during controlled exposure experiments, and the perceived received levels (RL, mean ± SD) during calibrations in 14 m water depth. The SL of the

research vessel was 140 ± 2 dB re 1µPa @1 m. Broadband ambient noise was 103 dB re 1 µPa (18 August 2018).

- Transparent reporting form

## Data availability

The data has been uploaded to Dryad.

The following dataset was generated:

| Author(s) | Year | Dataset title | Dataset URL | Database and Identifier |
|---|---|---|---|---|
| Sprogis KR, Videsen S, Madsen P | 2020 | Data from: Vessel noise levels drive behavioural responses of humpback whales with implications for whale-watching | https://doi.org/10.5061/dryad.9kd51c5dq | Dryad Digital Repository, 10.5061/dryad.9kd51c5dq |

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
