## [Decision Letter]

**Acceptance summary:**

This rigorous study improves our understanding of how humpback whales respond to boats, by showing that whales are sensitive to vessel noise level (dB). Using a playback experiment, the study shows that high noise levels lead whales to reduce the proportion of time resting, double their respiration rate and increase swim speed, relative to lower noise levels. The management implication is an important one: that good whale-watching practice should consider vessel noise levels, not only vessel proximity.

**Decision letter after peer review:**

Thank you for submitting your article "Vessel noise levels drive behavioural responses of humpback whales with implications for whale-watching" for consideration by *eLife*. Your article has been reviewed by three peer reviewers, and the evaluation has been overseen by a Reviewing Editor and Christian Rutz as the Senior Editor. The following individual involved in the review of your submission has agreed to reveal their identity: Patrick Miller (Reviewer #1).

The reviewers have discussed their reviews with one another, and the Reviewing Editor has drafted this decision letter to help you prepare a revised submission.

Summary:

This study makes a valuable contribution to our understanding of whale behaviour in the context of whale-watching, by showing that vessel noise level (dB) is a key variable to which whales respond. Using a playback experiment, the authors find that high noise levels lead whales to reduce the proportion of time resting, double respiration rate and increase swim speed, relative to lower noise levels. The management implication is an important one: that vessel noise levels, not only vessel proximity, must be considered when seeking to reduce the disturbance of boats to whales. The empirical methods used in the study are rigorous, and the use of observation and response scoring via overhead video recording is also an important methodological advance. Replicating findings with noise stimuli generated from different vessel engine types would be useful, and further research is needed to understand if geometric effects might matter independent of how they affect noise levels.

Revisions:

1) References throughout the manuscript to "the primary driver" should be changed to "a driver" or "a sufficient driver".

A study can only show that one of two possible stimuli is the “primary” driver if it assesses both stimuli across a range of exposure levels and finds that one has a strong effect at a certain received level, and the other not. The control treatment (against which all statistical comparisons were made) simulated a vessel pass with a consistent geometry at 100 m distance at a slow speed of 1.5 knots, and no variations of distance, angle or speed were explored. The study therefore was not suited to compare fully and fairly these two stimuli options. What would be the effect of a very quiet but fast boat or jet ski passing close to whales? We don't know, but it could easily be more extreme than the response to a distant noisy boat. This revision is important because there is a risk that your results could be over-generalized and management of this issue could be complicated if noise alone is named the problem, when proximity, speed and angle could also still be very important factors that need further study. Likewise, social context can play an important role in triggering behavioral reactions, and the presence of active singers in the area needs to be considered.

2) The Discussion should include some acknowledgement that geometric position of boats can potentially impact whales, and so management of those aspects should be continued, with future research needed to test if/how they might drive responses independently of how they influence noise level. In particular, Dunlop et al., 2017; 2018, showed that behavioral reactions of migrating humpback whales are triggered by a combination of received sound level and proximity (no visual cues).

In the “Implications for management” and “Conclusion sections”, you are fully justified to use your study as a basis for including noise emission and exposure to whales as an important guideline. Speed and distance are described in the context of how they affect received noise levels, appropriately. We urge you here to reemphasize that proximity and geometry alone may still be important drivers of behavioural impacts independent of noise levels. Indeed, you still recommend retaining here the “distance, angle, speed” aspects of management rules, and these should be justified or managers might focus only on noise before further tests are done. Further research is needed to understand if these geometric effects might matter independent of how they affect noise level, which should be stated here as a further research recommendation.

3) Consider condensing the Materials and methods. Likewise, some sections of the Results could be presented in a table and the Discussion contains too much repetition of results. Some streamlining will improve the flow of the manuscript and make sure the reader focuses on the salient points.

4) Consider including in the Discussion the difference in spectral composition of playback noise compared to recorded (true) vessel noise (omission of frequencies <200 Hz).

---

## [Author Response]

1) References throughout the manuscript to "the primary driver" should be changed to "a driver" or "a sufficient driver".A study can only show that one of two possible stimuli is the “primary” driver if it assesses both stimuli across a range of exposure levels and finds that one has a strong effect at a certain received level, and the other not. The control treatment (against which all statistical comparisons were made) simulated a vessel pass with a consistent geometry at 100 m distance at a slow speed of 1.5 knots, and no variations of distance, angle or speed were explored. The study therefore was not suited to compare fully and fairly these two stimuli options. What would be the effect of a very quiet but fast boat or jet ski passing close to whales? We don't know, but it could easily be more extreme than the response to a distant noisy boat. This revision is important because there is a risk that your results could be over-generalized and management of this issue could be complicated if noise alone is named the problem, when proximity, speed and angle could also still be very important factors that need further study. Likewise, social context can play an important role in triggering behavioral reactions, and the presence of active singers in the area needs to be considered.

Good point, this has been updated throughout the manuscript to “a driver”, “sufficient driver” or “adequate driver/stimulus”.

2) The discussion should include some acknowledgement that geometric position of boats can potentially impact whales, and so management of those aspects should be continued, with future research needed to test if/how they might drive responses independently of how they influence noise level. In particular, Dunlop et al., 2017; 2018, showed that behavioral reactions of migrating humpback whales are triggered by a combination of received sound level and proximity (no visual cues).In the “Implications for management” and “Conclusion” sections, you are fully justified to use your study as a basis for including noise emission and exposure to whales as an important guideline. Speed and distance are described in the context of how they affect received noise levels, appropriately. We urge you here to reemphasize that proximity and geometry alone may still be important drivers of behavioural impacts independent of noise levels. Indeed, you still recommend retaining here the “distance, angle, speed” aspects of management rules, and these should be justified or managers might focus only on noise before further tests are done. Further research is needed to understand if these geometric effects might matter independent of how they affect noise level, which should be stated here as a further research recommendation.

Thank you, we fully agree and have added some sentences to highlight these points.

Whale-watching guidelines globally generally already account for angle of approach (geometric position), speed and distance. Noise emission standards have not been incorporated into whale-watching guidelines. During whale-watching the angle of approach to whales is generally from the rear or side, as mentioned in the Introduction: “By stipulating approach distance (typically ~100 m), angle (typically from the rear and side), speed (typically below wake speed), current guidelines rest on the premise that physical proximity is the primary vehicle of disturbance (Higham et al., 2014). As such, a very quiet whale-watch vessel is considered to have the same impact on the target animals as a very noisy vessel at the same distance, angle and speed”. In the Discussion, we mentioned that: “Thus, to facilitate sustainable whale-watching operations, guidelines and legislation should be based on approach distance, angle, speed and noise emission standards”. We have now added some text in this Discussion paragraph to elaborate that approach angle (i.e. the placement of underwater noise) is of importance to examine further: “To further examine the impacts of vessel noise, we argue that future research should the address the noise effects from the presence of multiple vessels, different vessel engine types, vessel proximity and different vessel approach types (e.g. in-path vs. parallel approaches during whale-watching/swim-with-cetacean tourism)”. In the conclusions, we have elaborated in a sentence to ensure regulators keep their current guidelines/regulations, and add vessel noise to current guidelines, rather than only opting for vessel noise emission standards: “To reduce or avoid disturbance, we propose that noise emission standards be incorporated into whale-watch and swim-with-cetacean guidelines along with the already stipulated speed, distance and angle of approach regulations”.

3) Consider condensing the Materials and methods. Likewise, some sections of the Results could be presented in a table and the Discussion contains too much repetition of results. Some streamlining will improve the flow of the manuscript and make sure the reader focuses on the salient points.

Thank you, we agree and have shortened the Materials and methods by removing unnecessary words and several sections, for example:

– We changed from bullet points to a sentence: “(*before* phase = absence of vessel, stationary >300-400 m from whales with engine in neutral; *during* phase = vessel approach (Figure 1); *after* phase = departure of vessel, stationary >300 m from whales with engine in neutral). In the *during* phase, the driver of the vessel aimed for a tangential to parallel approach and pass at the same distance (~100 m) distance and speed (~1.5 knts)”.

– We shortened sentences like: “To ensure samples were independent, photo-identification of the dorsal fin using a DSLR (Canon 50D 400 mm lens) and aerial photographs of the dorsal side were taken using an unmanned aerial vehicle (UAV), to ensure the same mother-calf pair was never sampled twice.”

The results were also shortened by removing unnecessary/repeated phrases or sentences (thus a table was not needed as it would not have cut down in space).

In the Discussion section, to cut down on words and not repeat the results so much, we altered this sentence to remove several sentences: “Furthermore, the severity of behavioural responses differed among treatments, with louder noise exposures in *during* phases causing more severe behavioural responses in mothers (Figure 4)”. We also removed the sentence “To do that, we played back control/low, medium and high vessel noise treatments mimicking a relevant range of LF-weighted SLs of 124/148, 160 and 172 dB re 1 μPa at 100 m distance”. Some of the results on effect size were left in to allow the reader to easily find the key results and understand the magnitude of the responses which we feel is of importance.

4) Consider including in the Discussion the difference in spectral composition of playback noise compared to recorded (true) vessel noise (omission of frequencies <200 Hz).

Because the exposed whales are logging at the surface, their lower jaw will be about 1m below the surface. That means that frequencies with wavelengths that are four times longer than that depth will cancel out via interference with the surface (Lloyd’s mirror effect). For a 1 meter receiver depth, this implies cancellation of sound energy below some (1500m/s/6m) = 250 Hz. Thus, despite that our speaker setup cannot efficiently produce the needed sound energy below 200 Hz to fully match the recorded vessel noise signature, energy at these frequencies will never be part of the received noise for a whale logging at the surface. We have made that point in the revised Materials and methods. Also, we have included in the Materials and methods a reference to the power spectra and spectrograms of the recorded and playback noise added to the supplementary material (Figure 1—figure supplement 3) to demonstrate the slightly more tonal components in the playback noise. The sentence in the Materials and method states: “Figure 1—figure supplement 3 shows the spectral components of the playback sounds compared to the original moored vessel recording (6 m, 5 knts). It is shown that the playback noise is slightly more tonal than the recorded noise, but well within the spectral realm of whale-watching vessel spectral signatures (Figure 1—figure supplement 2). There is less energy at frequencies below 200 Hz in the playback noise compared to the recorded noise, however because the noise was played to whales logging at the surface, energy at frequencies below 200-400 Hz cancel out via interference with the surface and therefore does not reach the whale (Figure 1—figure supplement 3)”.